# Transformer Discharge Carbon-Trace Detection Based on Improved MSRCR Image-Enhancement Algorithm and YOLOv8 Model

**DOI:** 10.3390/s24134309

**Published:** 2024-07-02

**Authors:** Hongxin Ji, Peilin Han, Jiaqi Li, Xinghua Liu, Liqing Liu

**Affiliations:** 1School of Electrical Engineering, China University of Mining and Technology, Xuzhou 221116, China; ts22230107p31@cumt.edu.cn (P.H.); ts23230113p31@cumt.edu.cn (J.L.); 2College of Mechanical and Electronic Engineering, Shandong Agricultural University, Tai’an 271018, China; lxh9357@163.com; 3State Grid Tianjin Electric Power Research Institute, Tianjin 300180, China; liulq328@126.com

**Keywords:** transformer, discharge carbon trace, MSRCR, YOLOv8 model, SimAM, DyHead, pruning

## Abstract

It is difficult to visually detect internal defects in a large transformer with a metal closure. For convenient internal inspection, a micro-robot was adopted, and an inspection method based on an image-enhancement algorithm and an improved deep-learning network was proposed in this paper. Considering the dim environment inside the transformer and the problems of irregular imaging distance and fluctuating supplementary light conditions during image acquisition with the internal-inspection robot, an improved MSRCR algorithm for image enhancement was proposed. It could analyze the local contrast of the image and enhance the details on multiple scales. At the same time, a white-balance algorithm was introduced to enhance the contrast and brightness and solve the problems of overexposure and color distortion. To improve the target recognition performance of complex carbon-trace defects, the SimAM mechanism was incorporated into the Backbone network of the YOLOv8 model to enhance the extraction of carbon-trace features. Meanwhile, the DyHead dynamic detection Head framework was constructed at the output of the YOLOv8 model to improve the perception of local carbon traces with different sizes. To improve the defect target recognition speed of the transformer-inspection robot, a pruning operation was carried out on the YOLOv8 model to remove redundant parameters, realize model lightness, and improve detection efficiency. To verify the effectiveness of the improved algorithm, the detection model was trained and validated with the carbon-trace dataset. The results showed that the MSH-YOLOv8 algorithm achieved an accuracy of 91.80%, which was 3.4 percentage points higher compared to the original YOLOv8 algorithm, and had a significant advantage over other mainstream target-detection algorithms. Meanwhile, the FPS of the proposed algorithm was up to 99.2, indicating that the model computation and model complexity were successfully reduced, which meets the requirements for engineering applications of the transformer internal-inspection robot.

## 1. Introduction

As the core equipment of the power grid, the operating condition of large transformers is directly related to the stable operation of the power grid. The metal-enclosed nature of the transformer leads to poor internal visibility, and it is difficult to accurately determine the location, type, and severity of internal defects in the transformer only by analyzing the dissolved gases in the oil (e.g., the three-ratio method, the modified three-ratio method, etc.) [1,2,3,4,5,6,7,8]. To determine the internal defects of the transformer, manual drilling into the transformer or lifting the cover is often used for inspection, but this method has the problems of low efficiency, poor accuracy, high risk, and high cost. With the development of robotics and artificial intelligence, micro-robots can be utilized to enter the interior of large oil-immersed transformers without destroying the internal structure of the transformer to perform visual inspection and determine the location of insulation defects more intuitively.

At present, in the process of mobile inspection in transformer oil, the transformer internal-inspection robot uses vision cameras to continuously photograph the internal environment, wirelessly transmit the image data to an external computer and then relies on naked-eye observation to determine whether there are target defects on the photographed images. The large size of the actual large transformer results in many images being taken during inspection. The workload of visual judgment is huge, and it is easy to miss some target defects. The use of machine vision to automatically identify target defects and classify defect types is an important way to significantly improve efficiency. The detection accuracy of the transformer-inspection robot in the process of insulation defect detection depends on the extraction accuracy of the target features.

Early target-detection algorithms were mainly based on traditional computer vision techniques such as feature-based methods and machine-learning models. Representative approaches include the Viola–Jones detector based on Haar features [9] and linear classifiers based on Histogram of Oriented Gradients (HOG) features [10]. However, these methods usually perform poorly in terms of detection performance and robustness, especially in complex scenes and under changing light conditions. In recent years, with the rapid development of deep-learning techniques, deep neural network-based target-detection algorithms have achieved great success. In particular, the emergence of convolutional neural networks (CNNs) has revolutionized target detection [11]. First, the R-CNN family of methods was proposed, such as region-based CNN (R-CNN) [12], Fast R-CNN [13], and Faster R-CNN [14], which realize target detection through candidate region extraction and classification. Subsequently, single-stage detectors, such as You Only Look Once (YOLO) [15] and Single Shot MultiBox Detector (SSD) [16], further simplify the target-detection process and achieve real-time detection while making significant progress in accuracy. Among them, the YOLO algorithm and its derivative versions represent a breakthrough in the field of target detection. Its unique algorithmic structure not only achieves high-precision detection but also dramatically improves the processing speed, which is especially important in real-time application scenarios.

Due to the constraints of the fully enclosed metal shell of large transformers, the internal-inspection robot is easily affected by the environment and image-acquisition equipment when taking pictures of defects inside the transformer, and there are problems such as too strong or insufficient fill light, blockage, low contrast, motion blur and tilted shooting perspective, which leads to large differences in the quality of the acquired images, blurred images, inconspicuous carbon-trace feature, and other defects. Although the constantly updated YOLO series of target recognition algorithms have already achieved good performance, there are still problems, such as missed detection and false detection when facing the complex background inside the transformer and the transformer defects with complex features, different sizes, and irregularities. The main contributions of this paper include three aspects: the improvement of image enhancement, deep-learning network model improvement, and model pruning.


When the moving internal-inspection robot acquires images in the dark environment inside the transformer, there are unfavorable conditions such as irregular imaging distance and fluctuating supplemental light conditions, which lead to the problems of large differences in image quality, blurred images, and inconspicuous carbon-trace features. In this paper, an improved MSRCR image-enhancement algorithm is proposed. By enhancing the contrast of the image at multiple scales, highlighting the image details and structure, and fusing the multiple-scale components according to a certain weighting while introducing a white-balance algorithm to improve the brightness overexposure and color distortion problems, the brightness and contrast of the picture are enhanced.To improve the performance of complex carbon-trace target recognition, this paper incorporates the SimAM module into the Backbone network of the YOLOv8 deep-learning network model to enhance the ability to extract carbon-trace features. At the same time, the DyHead dynamic detection head framework is constructed at the output of the YOLOv8 model to enhance the ability to sense localized carbon traces with different sizes and further improve the accuracy and robustness of defect detection.To improve the recognition speed of the fast-moving transformer internal-inspection robot, this paper performs a pruning operation on the YOLOv8 deep-learning network model, removing redundant parameters and realizing model lightness. Through pruning optimization, the improved network significantly improves detection efficiency and meets the real-time requirements of engineering applications.


## 2. Structure of the Transformer Internal-Inspection Robot

The inspection robot mainly includes the main body, ultrasonic emission module, image-acquisition device, ultrasonic ranging module, vertical propeller propulsion device, horizontal propeller propulsion device, pressure sensor, and robot control system. The main body of the robot is an elliptical sealed structure, and the ultrasonic emission module is installed at the top of the main body, which is mainly used for the three-dimensional positioning of the inspection robot. An image-acquisition device is installed on the upper part of the robot body, which can be used to inspect the insulation condition inside the transformer as well as the surrounding obstacles. Four ultrasonic ranging modules are installed around the robot body, which can be used to measure the distance of the surrounding obstacles. Two vertical propeller thrusters are installed in the middle and lower part of the robot body, which can control the upward and downward movement of the robot body. Two horizontal propeller propulsion devices are installed in the lower part of the robot body, which can control the forward movement and rotation of the inspection robot body. A depth pressure sensor is installed at the bottom of the robot body, which can be used to measure the diving depth of the inspection robot. There is a control system inside the robot body, which mainly includes a motor drive and control module, data acquisition module, position detection module, wireless transmission module, and power-supply module. Its structure is shown in Figure 1.

The prototype of the transformer-inspection robot is made of aluminum alloy. The sealing and movement performance of the robot were tested in the previous period. In addition, it has initially realized the functions of floating up and sinking down, horizontal 360-degree rotation, forward and backward in the transformer oil.

During the moving process in the transformer oil, the image-acquisition module of the robot continuously captures the internal environment, and the collected internal image data are wirelessly transmitted to the external computer, thus completing the visual identification of transformer defects.

## 3. Sample Library Construction and Image Preprocessing of Carbon-Trace Defects

### 3.1. Sample Library Construction of Typical Defects

Because the types of insulation defects inside the actual transformer are abundant and the samples used for the training of the recognition model are scarce, this paper utilized the indoor test to reproduce the on-site defects, enriching the sample library of insulation defects inside the transformer and facilitating the training of the deep-learning-based target recognition model constructed in this paper.

The sample library constructed in this paper was the discharge carbon-trace defects, which are generated with partial discharge occurring on the surface of the transformer insulation enclosure, including dendritic carbon-trace defects and agglomerated creepage carbon-trace defects. A needle-plate model test platform was constructed in this paper to construct the carbon-trace sample library, which mainly included a specimen model, a boosting platform, and a high-speed camera. The specimen model was mainly composed of nylon screw, acrylic board, nylon bracket, front electrode, equalizing ring, and connecting rod. Its structure is shown in Figure 2. The transformer cardboard was uniformly cut to the size of 25 cm × 15 cm, and the nylon bracket was used to fix the cardboard on the acrylic board and the inclination angle of the cardboard was changed by replacing the bracket with different angles. The test transformer was SB-10KVA/100KV, and the sample container was made of acrylic, which was transparent and made it easy to observe the test phenomenon. The transformer oil used for the test was Kelamayi No. 25 transformer oil. The high-speed camera was HTSUA134GC/M, with 1.3 million pixels and a frame rate of 211FPS. Its structure is shown in Figure 3.

The partial discharge was generated using the test setup described above, and the camera was used to photograph, record, and upload the test phenomena to the computer. After many tests, a sample library of carbon-trace defects in the transformer enclosure screen was produced. As shown in Figure 4.

### 3.2. Preprocessing for Image Feature Enhancement

In the closed internal environment of the transformer, ambient light, noise, and other disturbing factors lead to poor-quality images captured by the internal-inspection robot and the inconspicuousness of carbon-trace features. This would cause trouble with subsequent image processing and feature extraction, reducing the visibility and identification accuracy of fault defects.

Image preprocessing was performed using the MSRCR algorithm to address the problems associated with the local carbon-trace images. The MSRCR algorithm is an image-enhancement method based on the Retinex theory, which performs localized contrast enhancement of an image while ensuring that the global contrast of the image remains unchanged, thus avoiding over-sharpening and enhancing the image brightness at the same time [17]. An image is represented as the product of the illumination component and the reflection component based on the Retinex theory [18]:(1)I(x,y)=L(x,y)R(x,y)
where I(x,y) denotes the original image, L(x,y) is the illuminance component and R(x,y) is the reflectance component. Retinex theory suggests that the color perceived by the human visual system depends mainly on the reflection component. For an original image I(x,y), the calculated result R(x,y) is the enhanced image. The illuminance component L(x,y) can be obtained by Gaussian blurring the original data.

The MSRCR algorithm is based on the multi-scale Retinex image-enhancement algorithm by introducing a color recovery factor, which compensates for the color distortion caused by image enhancement as well as the problem of grayish tones, whose expression is:(2)$$R_{MSRCR_{i}}(x,y)=C_{i}(x,y)\sum_{n=1}^{N}\omega_{n}\left\{\mathrm{lg}I_{i}(x,y)-\mathrm{lg}[F_{n}(x,y)*I_{i}(x,y)]\right\}$$
where $I_{i}(x,y)$ is the input image, $\omega_{n}$ is the weighting coefficient of the *n*th scale; $F_{n}(x,y)$ is the Gaussian filter function at the *n*th scale. $C_{i}(x,y)$ is the color recovery factor, whose expression is
(3)$$C_{i}(x,y)=\beta \left\{\mathrm{lg}[\alpha I_{i}(x,y)+1]-\mathrm{lg}[\sum_{i=1}^{3}I_{i}(x,y)+1]\right\}$$
where β is the gain function set to 5, and α is the non-linear strength set to 0.5.

Test results revealed that the images enhanced by the MSRCR algorithm suffered from overexposure and color distortion. To solve this problem, an improved MSRCR image-enhancement algorithm incorporating white balance was proposed in this paper. The principle of white balance is based on the grayscale world assumption and luminance invariance assumption. The grayscale world assumption holds that the average grayscale of all colors in an image should be equal under average illumination. The luminance invariance assumption holds that the luminance of an image should remain constant under different lighting conditions. Based on these assumptions, the goal of white balance is to make the average gray value of each channel in the image equal to eliminate the color bias [19], and the calculation formula is shown below.
(4)$$\mathrm{white}\text{\_}\mathrm{balance}=(\frac{\max\text{\_}avg^{power}}{channel\text{\_}avg^{power}})$$
where white_balance is the white-balance coefficient, max_avg is the maximum value of the average brightness of all channels in the image, channel_avg is the average brightness of each channel in the image. According to the white-balance coefficient, readjust the RGB channel values to solve the problem of brightness overexposure and color distortion.

As shown in Figure 5, the quantization results of the images before and after the enhancement in terms of brightness and contrast were presented. The brightness of the image enhanced by the MSRCR algorithm was concentrated around 170, which was too high and led to color distortion of the carbon-trace features and affected the image contrast. After the introduction of the improved MSRCR algorithm processing of white balance, the brightness of the image was improved, and the brightness value was concentrated at about 100 and uniformly distributed, which effectively improved the problem of overexposure and color distortion caused by too high brightness in the image after enhancement by the MSRCR algorithm. The histogram showed that the grayscale values of the improved image were more evenly distributed, indicating that the contrast has been significantly improved compared to the original algorithm.

## 4. The Improved Target-Detection Model Based on YOLOv8

Yolov8 is an updated version of the Yolo series algorithms, which has been widely used in the field of visual target detection. Yolov8 mainly consists of three parts: Backbone, Neck, and Head. The Backbone network is used to extract defective features for use by the subsequent networks. The Neck network fuses features extracted from different levels in the Backbone network to extract higher-level features. The Head network makes predictions based on the output features of the front network to obtain the final prediction results. The improved YOLOv8 network structure is shown in Figure 6.

After the carbon-trace image is input into the network architecture, the carbon-trace defect features, such as texture, shape, and color of the defects, are first extracted by several modules consisting of convolution and pooling layers; then the extracted defect features are fused and enhanced using the feature pyramid module to obtain more comprehensive defect information. Then, the above defect information will be transferred to the header network to perform classification and prediction, locate the defect positions, and classify the defect types. Finally, after the post-processing operation, the detection result is optimized to ensure that each defect is complete and only once labeled.

The existing Yolov8 algorithm has problems with leakage and false detection in the face of the complex background inside the transformer and the complexity of the carbon-trace features in the transformer, which are of various sizes and without any regularity. To solve this problem and further improve the defect detection efficiency of the improved model, this paper mainly focused on the improvement of three aspects: attention mechanism, target-detection Head, and model pruning.

### 4.1. SimAM Attention Mechanism

To improve the ability to extract carbon-trace features, an attention mechanism is introduced into the Backbone network. Most of the existing attention modules focus on the channel or spatial domain, generate one- or two-dimensional weights, and treat neurons in each channel or spatial location equally, which limits the ability to learn more discriminative cues [20]. Inspired by the attention mechanism of the human brain, an attention model with full three-dimensional weights was proposed by combining the two mechanisms in the channel and spatial domains, and an energy function was designed to compute and assign the weights to each neuron.

SimAM evaluates the importance of individual neurons by measuring the linear separability between a target neuron and other neurons and defines the following energy function for each neuron:(5)$$e_{t}(\omega_{t},b_{t},y,x_{i})={\left(y_{t}-\hat{t}\right)}^{2}+\frac{1}{M-1}\sum_{i=1}^{M-1}{\left(y_{o}-{\hat{x}}_{i}\right)}^{2}$$
where $\hat{t}=\omega_{t}t+b_{t}$ and ${\hat{x}}_{i}=\omega_{t}x_{i}+b_{t}$ are linear transformations of t and $x_{i}$, and $X\in R^{C\times H\times W}$ is the input features of t and i, i is the index in the spatial dimension, M=H×W are the number of neurons in the channel, $\omega_{t}$ and $b_{t}$ are the weighted and biased transformations, respectively. When $\hat{t}$ and $y_{t}$ reach the minimum values at the same time, the above equation is equivalent to finding the linear separability between the target neuron *t* and all other neurons in the same channel.

Using binary labels (i.e., 1 and −1) for $y_{t}$ and $y_{o}$, introducing a regularization factor and assuming that the mean and variance can be computed on all neurons, reapply it to all neurons on that channel to obtain a minimum energy formula:(6)$$e_{t}^{*}=\frac{4({\hat{\sigma }}^{2}+\lambda )}{{\left(t-\hat{\mu }\right)}^{2}+2{\hat{\sigma }}^{2}+2\lambda }$$
where $\hat{\mu }=\frac{1}{M}\sum_{i=1}^{M}x_{i}$, ${\hat{\sigma }}^{2}=\frac{1}{M}{\sum_{i=1}^{M}\left(x_{i}-\hat{\mu }\right)}^{2}$.

The above equation shows that at lower energies, the neuron becomes more distinct from the peripheral neurons. Therefore, the importance of each neuron can be obtained with $1/e_{t}^{*}$. Finally, the final weight of each neuron is obtained by scaling:(7)$$\tilde{X}=Sigmoid(\frac{1}{E})⊙X$$
where X and $\tilde{X}$ denote the input and output feature maps, and E is an energy matrix containing the energy value of each element in the input feature map *X*. The energy inverse is compressed using the Sigmoid function to limit its value to between 0 and 1. This is to avoid excessively large energy values while maintaining the relative importance of each pixel or neuron. Its structure is shown in Figure 7.

### 4.2. Head Frame for Self-Attentive Dynamic Detection

A Backbone network responsible for feature extraction and a Head detection network responsible for localization and classification are common design ideas in current target-detection technologies. Therefore, enhancing the performance of the Head network becomes one of the keys to improving the accuracy of target detection. As an innovative, dynamic detection Head framework, the DyHead dramatically enhances the characterization capability of the detection Head by integrating three self-attention mechanisms with different dimensions (level, spatial, and channel), i.e., scale-awareness, space-awareness, and task-awareness mechanisms [21]. The scale-aware mechanism focuses on the level dimension and enhances the scale characterization of a specific layer by assessing the importance of different semantic dimensions, computed as:(8)$$\pi_{L}(F)\cdot F=\sigma (f(\frac{1}{SC}\sum_{S,C}F))\cdot F$$
where *F* represents the input feature map, *S* represents the spatial dimension, *C* represents the number of channels, and *σ* is a hard-sigmoid function to limit the attention weight range between 0 and 1.

The spatial perception mechanism is then deployed in the spatial dimension (height × width) and works to learn coherent and discriminative feature representations across spatial locations, computed as:(9)$$\pi_{S}(F)\cdot F=\frac{1}{L}\sum_{l=1}^{L}\sum_{k=1}^{K}\omega_{l,k}\cdot F(l;p_{k}+\Delta p_{k};c)\cdot \Delta m_{k}$$
where *K* is the number of sparsely sampled locations, $p_{k}+\Delta p_{k}$ is the self-learning spatial offset, and $\Delta m_{k}$ is a scalar of the self-learning importance at the location $p_{k}$.

The task-aware mechanism runs through the channel dimensions, guiding different feature channels to support their respective tasks (e.g., classification, regression, and center/keypoint detection). Based on the object-specific convolutional kernel response, it is computed as:(10)$$\pi_{C}(F)\cdot F=\max(\alpha^{1}(F)\cdot F_{c}+\beta^{1}(F),\alpha^{2}(F)\cdot F_{c}+\beta^{2}(F))$$
where ${\left[\alpha^{1},\alpha^{2},\beta^{1},\beta^{2}\right]}^{T}=\theta (\cdot )$ is the hyperfunction of the learning control activation threshold.

By decomposing the attentional process into three separate dimensions, each focusing on one perspective, DyHead achieves an effective unification of the target-detection Head and the attentional mechanism:(11)$$W(F)=\pi_{C}(\pi_{S}(\pi_{L}(F)\cdot F)\cdot F)\cdot F$$
where F is the three-dimensional feature tensor and $\pi_{L}(\cdot )$, $\pi_{S}(\cdot )$, $\pi_{C}(\cdot )$ is the attention function applied to the three dimensions L,S,C, respectively. The above three attention mechanisms are sequentially connected in the model line and repeatedly stacked to form the DyHead module, the structure of which is shown in Figure 8.

In this paper, the DyHead module was integrated into the detection Head network of Yolov8, which can be integrated into a target-detection Head by effectively combining multiple self-attentive attention mechanisms across feature layers, enabling it to learn the relative importance between feature layers and enhancing the feature-enhanced perception of localized carbon traces with different sizes on the corresponding layers.

### 4.3. Lightweight Pruning for Model Optimization 

To improve the detection speed of the improved model, make the model lighter, reduce overfitting, and enhance the interpretability of the model, a global pruning algorithm, LAMP, was adopted in this paper, which is based on the computation of importance scores [22]. The principle is to calculate the size of the target connection square weights and then normalize them by the sum of all the “surviving weights” in the layer. Suppose that the weight tensor of each fully connected or convolutional layer is expanded into a one-dimensional vector, and these weights are sorted by a given index term, where the LAMP score of the first index of the weight tensor is defined as:(12)$$score(u;W):=\frac{{\left(W[u]\right)}^{2}}{\sum_{v\geq u}{\left(W[v]\right)}^{2}}$$
where W denotes the weight, W[u] and W[v] are the weight terms mapped by indexes u and v, respectively, after ascending order.

The LAMP score is used to evaluate the relative importance of all surviving connections in the same layer. Once the score is determined, connections with smaller weight sizes (in the same layer) need to be pruned until the desired global sparsity constraint is satisfied. The pruning process is shown in Figure 9.

The specific process of the LAMP pruning algorithm is: (1) According to the improved yolov8 network architecture, train the network to obtain the weight file of the training results. (2) According to the weight file, the LAMP scores of each connection in each layer are calculated, and the connections with lower scores in each layer are selected to be pruned according to the preset global sparsity requirements. (3) Retrain the pruned model to restore the model to its pre-pruning performance. (4) Evaluate the performance of the retrained model.

## 5. Model Validation and Analysis

### 5.1. Calibration of Samples

Data expansion was performed on the carbon-trace samples obtained from the indoor experiment, and a total of 6253 images were obtained. The images were divided according to the training set:validation set:dataset ratio of 8:1:1 after data preprocessing, and the dataset was labeled using LabelImg.

### 5.2. Test Platform Parameters and Evaluation Indexes

The model-training environment is Windows 11, the CPU model is Intel Corei5-13400F, and the GPU model is NVIDIA GeForce 4060ti, with 16 GB of video memory. The deep-learning framework is PyTorch 2.1.0, and the Python version is 3.8. The training parameters are set as follows: stochastic gradient descent (SGD) is used as the optimizer, the initial learning rate is set to 0.01, and the number of training rounds is set to 200 epochs.

To evaluate the performance of the algorithm, it is necessary to select appropriate evaluation indices. In this paper, mean Average Precision (mAP) is selected as the evaluation index. The higher the mAP, the better the algorithm’s performance in both checking accuracy and checking completeness [23]. First, the precision (P) and recall (R) of the prediction frames of each category above the threshold of the intersection over the union (IoU) are calculated, respectively,
(13)$$P=\frac{TP}{TP+FP}$$
(14)$$R=\frac{TP}{TP+FN}$$
where TP denotes the number of correctly identified positive samples, FP denotes the number of negative samples identified as positive samples, FN denotes the number of positive samples incorrectly identified as negative samples, TN denotes the number of negative samples correctly identified. The calculation results of the checking accuracy and the checking completeness constitute the P–R curve.

For each category, its Average Precision (AP), which is the area below the P–R curve, is calculated [16]. For each category, the maximum value of AP is taken as the AP value for that category. Finally, the AP values of all categories were averaged to obtain the final mAP value.

### 5.3. Ablation Experiment

To further verify the effectiveness of the attention mechanism SimAM and the self-attention detection head network DyHead, ablation experiments were carried out with YOLOv8n as the benchmark. The optimal results in the training rounds are selected for comparison, and the test results are shown in Table 1, where √ indicates that the improved method is used.

As shown in Table 1, compared with the baseline model, the mAP of the improved models gradually increased. After the addition of the SimAM attention mechanism, the recall rate R increased by 0.025, and the mAP increased by 0.020; after the introduction of the DyHead self-attentive dynamic monitoring Head, the precision rate P increased by 0.015, the recall rate R increased by 0.019, and the mAP increased by 0.030. Finally, under the joint effect of SimAM and DyHead, the precision rate and the recall rate were both effectively improved, and mAP was improved by 0.037, in which the precision of dendritic carbon-trace recognition was improved by 0.051, and the precision of clumpy carbon-trace detection was improved by 0.024.

To verify the performance of the proposed algorithm in a real environment, this paper collected real images of large transformers after partial discharges occurred inside them for carbon-trace identification and localization. The results shown in Figure 10 illustrate the ability of the algorithm to effectively detect authentic carbon traces inside the transformers, even under complex background scenarios. This demonstrated the practical effectiveness of the algorithm proposed in this paper.

### 5.4. Comparative Experiments with Model Pruning

Since there are some layers in the network architecture whose data are very important for the whole network, pruning these layers will lose the relevant data and lead to a significant degradation of the model performance. Therefore, before pruning the model, the model needs to be processed by layer hopping. In this paper, layer hopping was applied to the output layer of the detection Head network.

Then, the improved model was pruned, and the results of pruning are shown in Table 2, where the parameter speed_up was used to set the degree of pruning. The larger the speed_up, the larger the degree of pruning of the model.

In Table 2, Parameters represents the number of model parameters, which reflect the computational size of the model; FLOPs indicates the complexity of the model. When speed_up was set to 2, the FLOPs of the model was reduced, and the computational amount was reduced by 65.2%, which significantly reduced the computational amount and complexity of the model. The FPS was also improved by 24.8%, and the mAP was only reduced by 0.002, which had a very small effect on the accuracy of the model. After increasing the speed_up parameter, the computation and parameter of the pruned model were further reduced, and the FPS was further improved. When speed_up was 2.5, the model improvement effect was the most balanced. The FPS was improved by 36.7%, the number of parameters was reduced by 71%, the FLOPs were reduced to 39.6% of the original, and the mAP value was only reduced by 0.003. Figure 11 demonstrates the comparison of each channel before and after pruning, with the orange color representing the channel that is not pruned and the red color representing the channel that is pruned. This indicates that there were many redundant parameters in the Backbone network. The pruning operation can remove the redundant parameters, reduce the model complexity and the risk of overfitting, and improve the interpretability of the model. The improvement of FPS is very important for real-time detection, and the pruned model could meet the requirements of real-time monitoring.

### 5.5. Comparative Experiments with Different Algorithms

To further validate the performance of the improved algorithm, it was compared with other target-detection algorithms, including YOLOv7-tiny [24], YOLOv6n [25], YOLOv5n, YOLOv3-tiny [26], YOLOv9t [27], YOLOv10n [28] under the same hardware and environment configuration. The detection results of different algorithms are shown in Table 3. Compared with other algorithms, MSH-YOLOv8 achieved better FLOPs, accuracy, and recall values with less computation consumption. Meanwhile, the mAP of the MSH-YOLOv8 was higher than YOLOv7-tiny, YOLOv6n, YOLOv5n, YOLOv3-tiny, YOLOv9, YOLOv10 with 0.203, 0.053, 0.146, 0.218, 0.002, and 0.042, respectively. Although the FPS of the proposed algorithm was not the highest, it could meet the requirement of real-time monitoring.

Figure 12 shows the detection results of different algorithms. YOLOv3-tiny, YOLOv5n, and YOLO7-tiny failed to detect all the carbon traces, reflecting comparatively lower detection accuracy. Although YOLOv6n and YOLOv10 could detect all the carbon traces, their detection accuracies were relatively low and may miss or misdetect when facing more complicated carbon traces.

To show the performance improvement of the models in this paper more clearly, the mAP curves of each model were plotted. As shown in Figure 13, compared with other models, the improved model MSH-YOLOv8 had a significant improvement in mAP. In summary, the model proposed in this paper had a significant improvement in performance.

## 6. Conclusions

(1) To improve the quality of the images suffering from the problems of darkness and fluctuation of supplementary light inside a transformer, an improved image-enhancement algorithm was proposed. It incorporated the white-balance algorithm into the MSRCR algorithm to enhance the brightness as well as the contrast of the captured images while overcoming the brightness overexposure and color distortion of the original algorithm.

(2) Aiming at the problem of misdetection and omission detection of carbon-trace defects under the complex transformer background, the novel MSH-YOLOv8 detection network was proposed. The SimAM attention mechanism was introduced into the Backbone of the YOLOv8 network, which improved the ability to extract the features of defective carbon traces of the transformers without introducing redundant computational parameters. The DyHead module was used to construct the framework of the self-attention dynamic detection Head. Compared to the original network, the proposed model achieved a 5.1% increase in precision, a 2.4% increase in recall, and a 3.7% increase in mAP, contributing to the improved recognition accuracy of carbon traces under the complex backgrounds inside the transformer.

(3) To enhance the recognition speed of the improved network and reduce the model parameters and complexity, the LAMP algorithm was utilized to perform pruning operations to reduce the redundant parameters in the model channels and enhance the detection efficiency of the model with minor loss of the model accuracy as much as possible. Finally, the number of parameters was reduced by 71%, and the FPS was improved by 36.7%, while the accuracy was only reduced by 0.3% to meet the requirements for transformer internal real-time monitoring requirements.

## Figures and Tables

**Figure 1 sensors-24-04309-f001:**
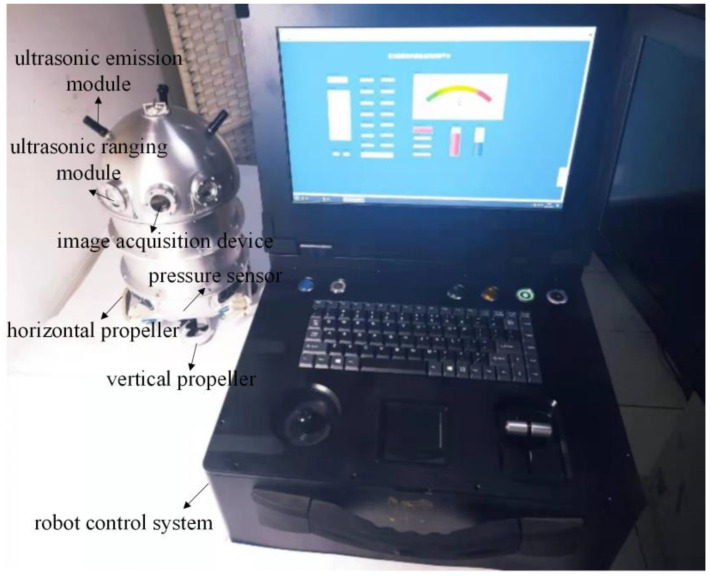
Physical diagram of transformer-inspection robot.

**Figure 2 sensors-24-04309-f002:**
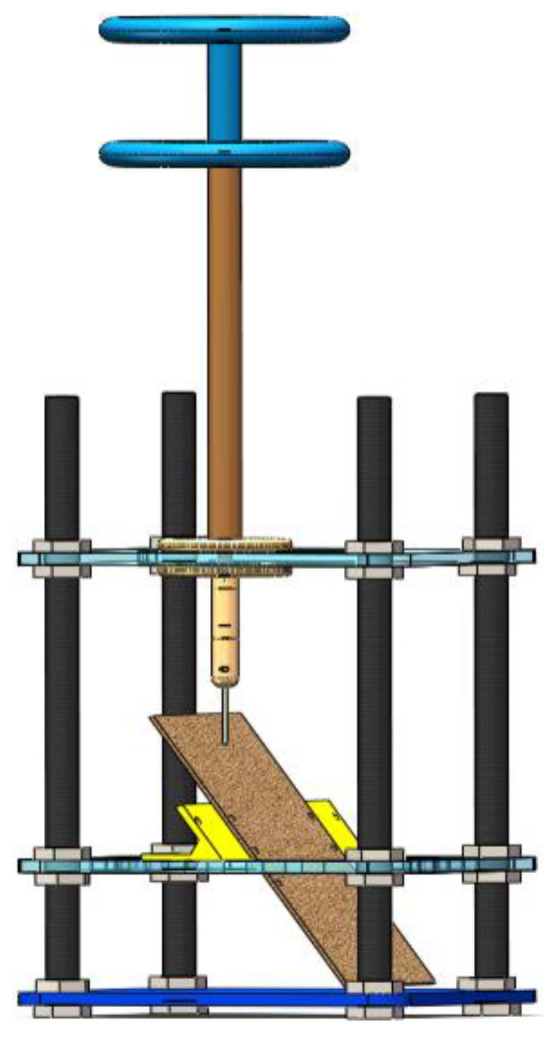
Needle-plate partial discharge model.

**Figure 3 sensors-24-04309-f003:**
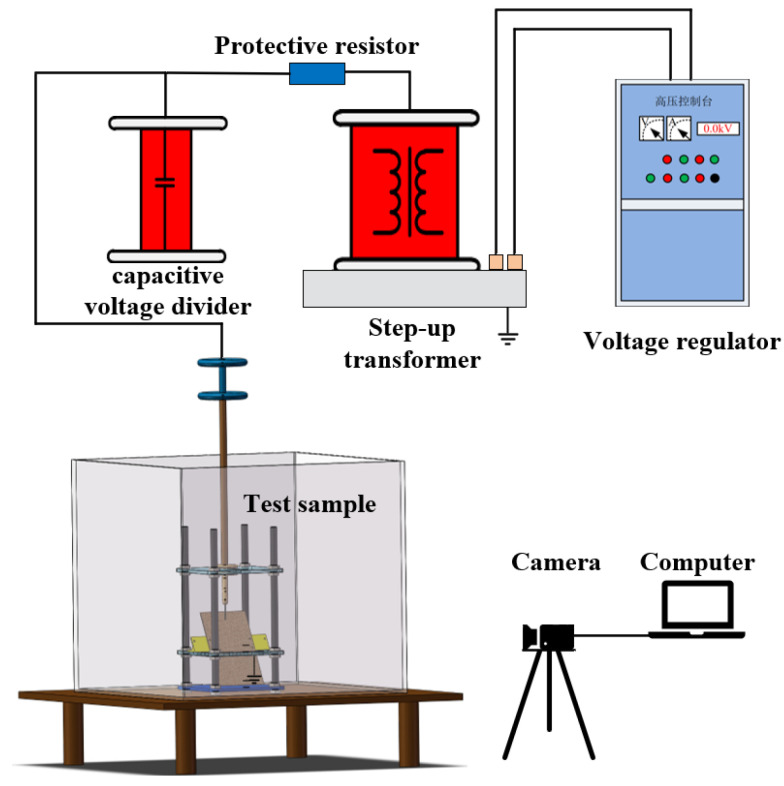
Schematic diagram for partial discharge (PD) test.

**Figure 4 sensors-24-04309-f004:**
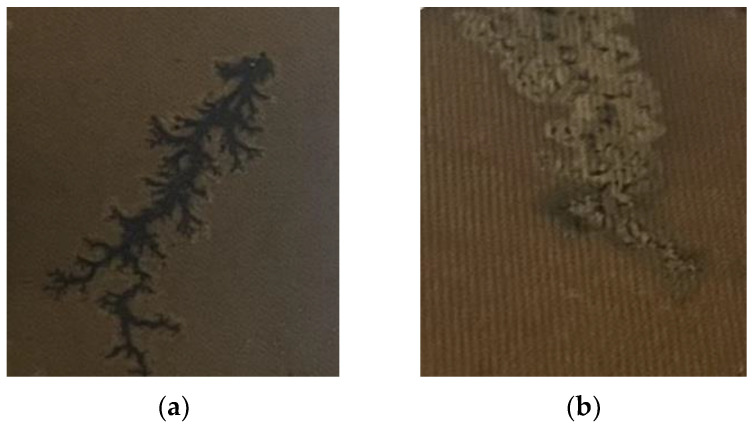
Carbon-trace images of partial discharge in the transformer: (**a**) dendritic; (**b**) clumpy.

**Figure 5 sensors-24-04309-f005:**
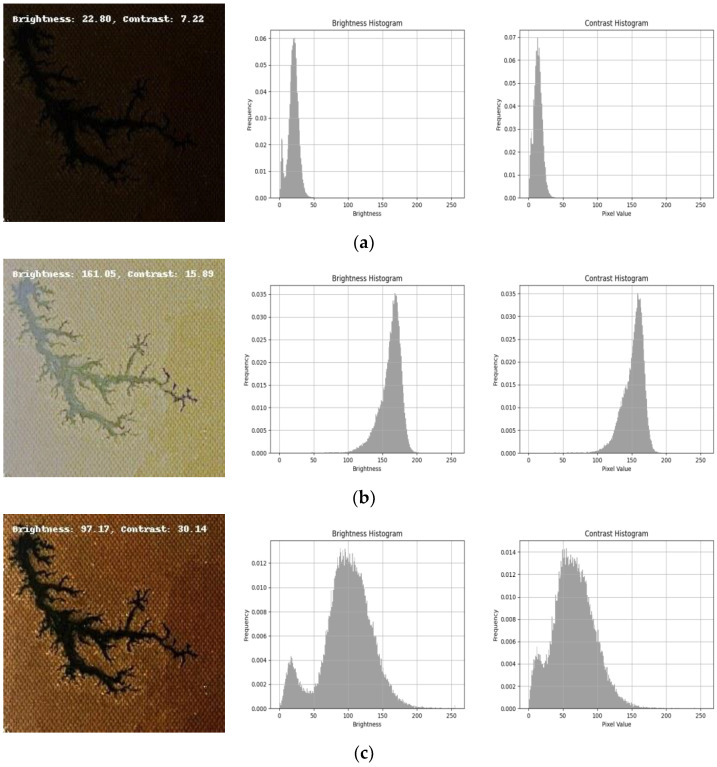
Different image-enhancement algorithm results: (**a**) original; (**b**) MSRCR; (**c**) improved MSRCR.

**Figure 6 sensors-24-04309-f006:**
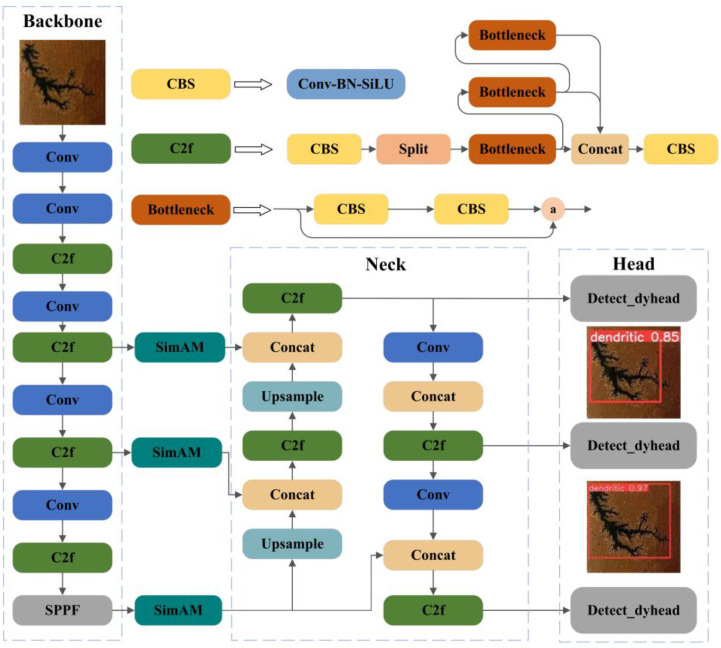
The structure of the improved YOLOv8 Network.

**Figure 7 sensors-24-04309-f007:**
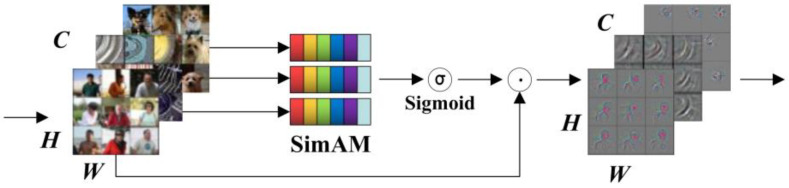
SimAM attention.

**Figure 8 sensors-24-04309-f008:**
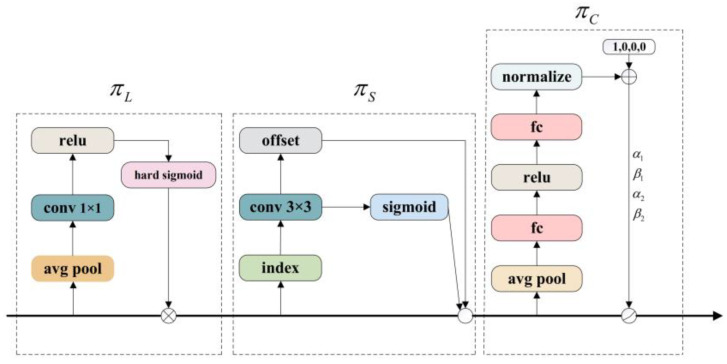
DyHead block.

**Figure 9 sensors-24-04309-f009:**
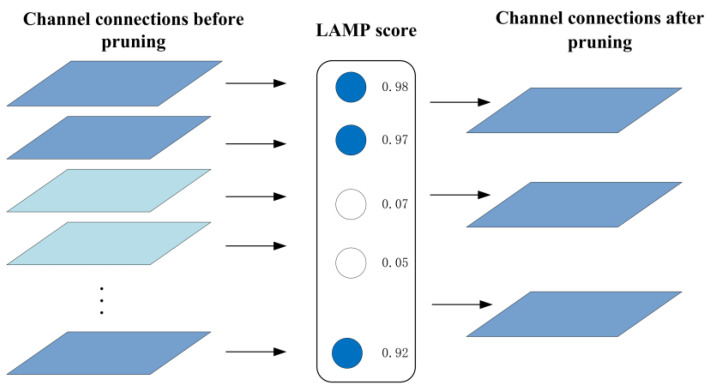
Schematic diagram of LAMP operation.

**Figure 10 sensors-24-04309-f010:**
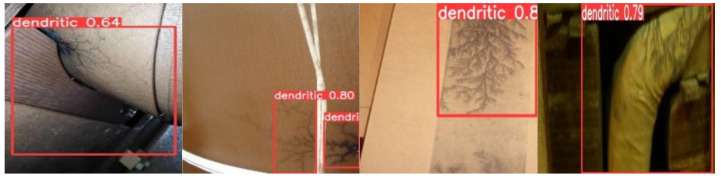
Detection performance of real transformer carbon traces.

**Figure 11 sensors-24-04309-f011:**
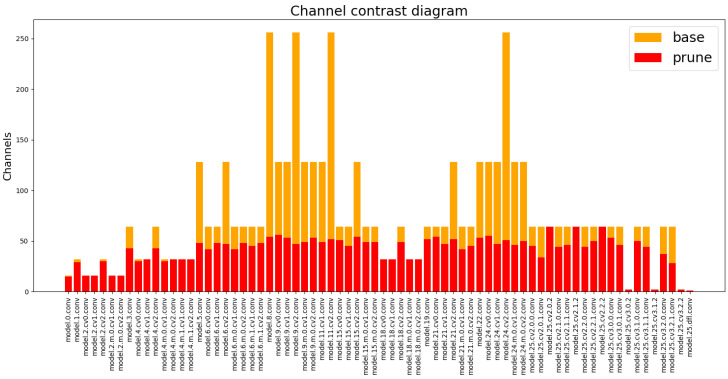
Comparison of channels before and after pruning.

**Figure 12 sensors-24-04309-f012:**
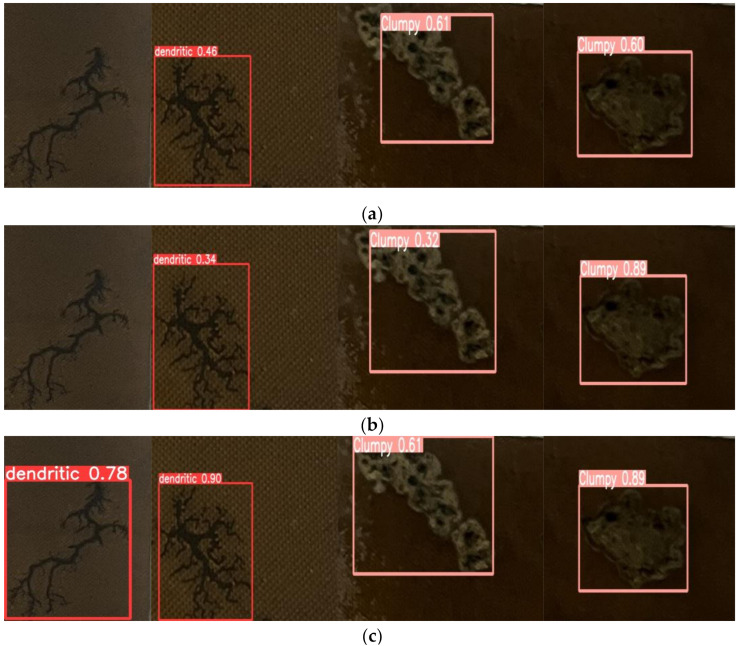
Detection results of different algorithms: (**a**) Yolov3-tiny; (**b**) Yolov5n; (**c**) Yolov6n; (**d**) Yolov7-tiny; (**e**) Yolov9t; (**f**)Yolov10n (**g**) MSH-Yolov8.

**Figure 13 sensors-24-04309-f013:**
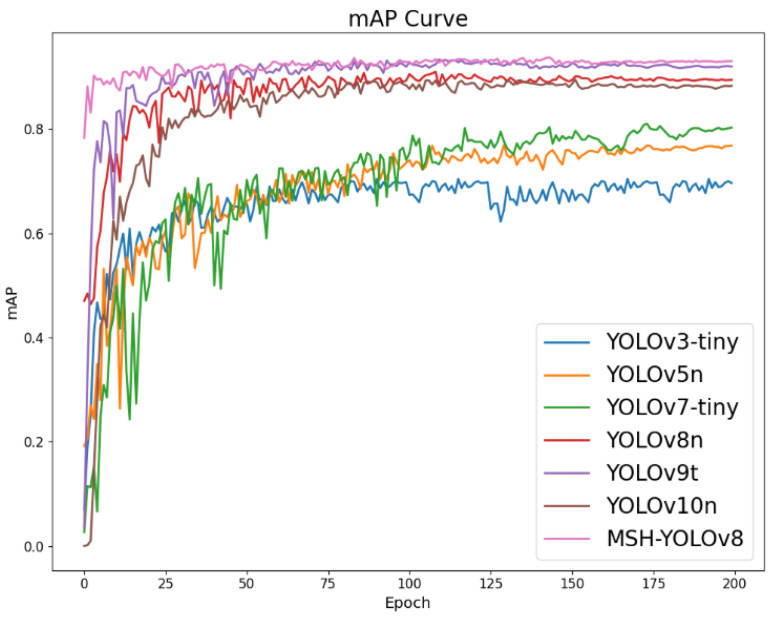
mAP Curves of different detection models.

**Table 1 sensors-24-04309-t001:** Results of ablation experiments.

No.	Baseline	SimAM	Dyhead	AP		P	R	mAP
				Dendritic	Clumpy			
1	√			0.881	0.886	0.949	0.828	0.884
2	√	√		0.919	0.889	0.948	0.853	0.904
3	√		√	0.925	0.902	0.964	0.847	0.914
4	√	√	√	0.932	0.910	0.967	0.852	0.921

**Table 2 sensors-24-04309-t002:** Comparison of model pruning effects.

Speed_Up	Parameters (M)	FLOPs (G)	FPS	mAP (%)
1	3.49	9.6	72.8	0.921
2	1.21	4.8	91.5	0.919
2.5	1.01	3.8	99.5	0.918
3	0.901	3.2	104.9	0.913

**Table 3 sensors-24-04309-t003:** Comparison of experimental results for different algorithms.

Network	Parameters (M)	FLOPs (G)	mAP (%)	FPS
YOLOv3-tiny	8.67	13.0	0.698	98
YOLOv5n	1.76	4.1	0.77	101
YOLOv6n	4.23	11.8	0.863	118
YOLOv7-tiny	6.01	13.0	0.775	52.9
YOLOv9t	2.53	11.2	0.916	45.7
YOLOv10n	2.69	8.2	0.876	93.4
MSH-YOLOv8	1.01	3.8	0.918	99.5

## Data Availability

The data presented in this study are available on request from the corresponding author. The data are not publicly available due to privacy.

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
