# Peer review of "Transformer Discharge Carbon-Trace Detection Based on Improved MSRCR Image-Enhancement Algorithm and YOLOv8 Model"

_sensors, 2024, doi:10.3390/s24134309_

Round 1
Reviewer 1 Report
Comments and Suggestions for Authors
An improved MSRCR Image Enhancement Algorithm combined with the YOLOv8 mode is proposed for transformer discharge carbon trace detection. The topic is pretty realisitic and the methodology is properly designed. However, there are some concerns regarding the quality of communications. Prior to publication, the following modifications are recommended.
1) Abbreviations, e.g. "MSRCR", "LAMP", when first used, must be accompanied by their full forms.
2) The quality of figure 1, where the transformer inspection robot is presented, should be improved. A zoom-in view of the robot with detailed annotations of the sensors, propeller propulsions, and the image acquisition device would be favorable.
3) The scales of fig. 5(b) and fig. 5(c) should be unified. In addition, it is not convincing to claim that "the improved MSRCR algorithm enhances the image brightness more naturally" . Define the term "naturally", or this conclusion should be justified by some quantitative index.
4) In equation (7), the operator is not properly defined.
5) In subsection 5.3, the symbol "[email protected]" is unclear.
6)In table 2, under speedup_factor=3, the parameters(M)=9.01. Is this correct?
7) Figure 11 must be updated. First, the fontsize of the annotations of the x-axis is tiny, providing little information. Secondly, the meaning and unit for the y-axis are not specified.
8) Most references have been published by 2022. The authors should relate their work to more recent advances in the area.
Comments on the Quality of English LanguageThe quality of english language is not satisfying. There are typos and inappropriate expressions throughout the manuscript. Some examples are listed as follows.
1) In the abstract section, maybe replacement of the phrase 'have large size differences' by 'multi-scale' can yield a more concise sentence.
2) Many equations are followed by a clause "where ..." for definition of symbols. Thus, "where" should not be capitalized, and the spacing should be removed.
3) In subsection 5.3, "are both are effectively improved", there's a redundant 'are'.
4) In the the first sentence of subsection 5.2, 'Inter Corei5-13400F' should be 'Intel Corei5-13400F'. Too many typos can weaken the scientific rigour of the manuscript, thus should be avoided.
5) In Conclusions, the comma at the end of the second paragraph should be replaced by a period. In addition, the spacing before the third paragraph is missing.
Author Response
"Please see the attachment.

Reviewer 2 Report
Comments and Suggestions for Authors
This article uses the MSRCR image enhancement algorithm and white balance to preprocess images, and constructs the MSH-YOLOv8 detection model for visual inspection of internal defects in transformers, which is innovative. However, further modifications are needed in the following aspects of the article.
1.The last paragraph of the Introduction should describe the contribution of this article's work item by item based on its importance.
2.The original image and legend of Figure 1 are displayed across pages. Please reconsider the layout of Figure 1.
3. This article constructs a typical defect sample library, and what are the differences and advantages compared to the currently publicly available dataset of internal insulation defects in transformers.
4.What are the values of gain function β and strength of the nonlinearity α mentioned in formula 3 in this article.
5. Please explain formula 8, such as what SC and F represent.
6. The abstract is too redundant, please streamline it.
Comments on the Quality of English LanguageThe English grammar needs to be modified.
Reviewer 3 Report
Comments and Suggestions for Authors
The paper fails to adequately highlight the novelty of its approach. While it claims to propose an improved image enhancement algorithm and target detection network, it lacks a clear comparison with existing methods to demonstrate its superiority. The paper does not sufficiently justify the choice of techniques used in the proposed method. For instance, it introduces the MSRCR image enhancement algorithm and the YOLOv8 model but does not explain why these specific choices were made or how they address the unique challenges of carbon trace detection in transformers.
The paper briefly mentions the practical effectiveness of the proposed algorithm but does not explore the potential challenges in deploying the system in actual transformer inspection scenarios. Without addressing these issues, it's challenging to assess the feasibility of the proposed approach in practical settings.
The paper does not provide information on accessing the source code or datasets used in the experiments, limiting reproducibility and obstructing other researchers from validating the results or building upon the proposed work.
The paper makes overly optimistic claims about the performance improvements achieved by the proposed method without sufficient evidence or comparative analysis. Claims of precision, recall, and FPS improvements should be backed by thorough experimentation and statistical analysis.
Round 2
Reviewer 1 Report
Comments and Suggestions for Authors
The authors have properly addressed all my previous concerns and made substantial corrections to the original manuscript. I recommend publication of this work in its current form.
Reviewer 3 Report
Comments and Suggestions for Authors
This revised version is suitable for publication.